# Isolation and Antimicrobial Sensitivity of *Mycoplasma synoviae* and *Mycoplasma gallisepticum* from Vaccinated Hens in Mexico

**DOI:** 10.3390/pathogens9110924

**Published:** 2020-11-07

**Authors:** Víctor M. Petrone-Garcia, Guillermo Tellez-Isaias, Fernando Alba-Hurtado, Christine N. Vuong, Raquel Lopez-Arellano

**Affiliations:** 1Programa de Doctorado en Ciencias de la Salud y Producción Animal, Facultad de Estudios Superiores Cuautitlán, Universidad Nacional Autónoma de México, Cuautitlán Izcalli 54714, Mexico; 2Department of Poultry Science, University of Arkansas, Fayetteville, AR 72704, USA; gtellez@uark.edu (G.T.-I.); vuong@uark.edu (C.N.V.); 3Departamento de Ciencias Biológicas, Facultad de Estudios Superiores Cuautitlán, Universidad Nacional Autónoma de México, Cuautitlán Izcalli 54714, Mexico; fealba@hotmail.com; 4Laboratorio No 5: LEDEFAR, Unidad de Investigación Multidisciplinaria, Facultad de Estudios Superiores Cuautitlán, Universidad Nacional Autónoma de México, Cuautitlán Izcalli 54714, Mexico; lopezar@unam.mx

**Keywords:** *Mycoplasma synoviae*, *curcumin*, thermosensitivity, pathogenicity, reversal, MS-H vaccine, tiamulin, tylosin

## Abstract

*Mycoplasma synoviae* (MS) and *Mycoplasma gallisepticum* (MG) strains were isolated at 39.5 °C to rule out temperature-sensitive strains (ts+) and identified using random amplification of polymorphic DNA. Then, their minimum inhibitory concentrations (MIC_100_) were calculated in isolated strains from broiler breeders and laying hens vaccinated with ts+ MS-H and ts+ MG TS-11 vaccines in Mexico. We sampled 631 lots of hens. A total of 28 of the 123 MS isolates and 12 of the 23 MG isolates were analyzed using random amplification of polymorphic DNA, of which 24 and 3 matched the DNA banding patterns of the MS-H and MG-F strains, respectively. The isolated MS and MG strains were sensitive to tiamulin and tylosin and showed intermediate sensitivity or resistance to lincomycin, florfenicol, erythromycin, enrofloxacin, and curcumin. Although both the MS and MG strains were sensitive to the same antibiotics (MIC_100_ lower than 1 mg mL^−1^), the MG strains were 5 to 10 times more sensitive than the MS strains. MS is the most frequently isolated mycoplasma in Mexican poultry production. The MS vaccine used (ts+ MS-H) could reverse its thermosensitivity and therefore could regain its virulence. MS was less sensitive to tiamulin and tylosin compared to MG.

## 1. Introduction

*Mycoplasma synoviae* (MS) and *Mycoplasma gallisepticum* (MG) are the most important *Mycoplasma* species for commercial poultry in Mexico. For decades, these species have been recognized as the cause of chronic respiratory disease (CRD) [1,2,3]. In addition, they decrease the fertile egg production in broiler breeders, cause late-stage embryonic death, or result in the births of infected chicks that later develop CRD. This disease suppresses the innate immune responses of the respiratory system and predisposes the bird to infection with *Escherichia coli*, producing complicated CRD (CCRD) [4]. CCRD is responsible for significant economic losses as it causes polyserositis, septicemia, and death in poultry farms as well as seizures at slaughterhouses [1,2].

In Mexico, mycoplasmosis is currently controlled by vaccination and antibiotic metaphylaxis. The only vaccine used in Mexico against MS is the MS-H strain. The MS-H strain is temperature sensitive (ts+) and was developed through chemical mutagenesis of an Australian field isolate (strain 86079/7NS) [5]. The ts+ MG vaccine strain used in Mexico is TS-11, which was also developed through chemical mutagenesis of an Australian field isolate (strain 80083) [6]. The most widely used groups of antibiotics against *Mycoplasma* spp. are macrolides, pleuromutilins, lincosamides, and fluoroquinolones. Other antibiotics with poor sensitivity, such as amphenicols and tetracyclines, are mainly used in combination to treat CCRD [7,8]. Because *Mycoplasma* spp. are primarily vertically transmitted from the hen to the chick, vaccination and treatments mainly focus on broiler breeders.

Considerable efforts have been made to develop more powerful, effective, well-tolerated, and, above all, safe medicines for humans. One of these medicines is curcumin (1, C21H20O6) or diferuloylmethane, which is extracted from the turmeric tuber (*Curcuma longa* L. Zingiberaceae) [9]. Studies on the in vitro antibacterial effect of curcumin on *Staphylococcus aureus*, *Staphylococcus intermedius*, *Staphylococcus epidermidis*, and *Edwardsiella tarda* [10] have demonstrated that the minimum inhibitory concentration (MIC) of curcumin solubilized in dimethyl sulfoxide against *Mycoplasma* spp. isolated from mammals ranges from 50 to 100 µg mL^−1^ [11], although its effect on mycoplasmas isolated from birds has not been demonstrated.

Random amplification of the polymorphic DNA (RAPD) identification method of MS and MG strains has proven efficient and particularly useful for epidemiological studies as well as the identification and differentiation of vaccine strains and field isolates [12,13,14]. A previous work [15] showed that RAPD had a discriminatory index for MS superior to 0.95; consequently, this molecular method was chosen for this study.

Eye or spray vaccination with ts+ MS and MG strains aim to colonize the upper respiratory tract and stimulate the local immune system without causing systemic colonization or transovarian transmission to the progeny. Reversing ts+ to ts− facilitates systemic colonization and transovarian transmission, predisposing the bird to CCRD [16]. Based on the above, the objective of the present study was to isolate and identify ts− vaccine strains from poultry vaccinated with ts+ strains using the RAPD method and evaluate the strains’ sensitivity to antimycoplasmic antibiotics.

## 2. Results

In the present study, 631 lots of hens were sampled from 14 poultry companies (Table 1). *Mycoplasma* spp. were isolated in 23.14% (146/631) of samples from 100% (14/14) of the poultry companies. Of the positive isolates, MS accounted for 84.25% (123/146) of the isolates and was significantly (*p* < 0.001) more frequently isolated than MG, which accounted for 15.75% (23/146) of isolates. Of the 123 MS isolates, 28 were analyzed using RAPD, 24 presented DNA banding patterns matching those of the MS-H strain, and four presented different DNA banding patterns. Of the 23 MG isolates, 12 were analyzed using RAPD, three presented DNA banding patterns matching those of the MG-F strain, and nine were untyped strains with DNA banding patterns different from those of the vaccine control strains (Table 2).

MS isolates were susceptible to tiamulin and tylosin but resistant to curcumin, erythromycin, and florfenicol. The MG isolates were susceptible to erythromycin, tiamulin, and tylosin (Table 3).

The isolates with DNA banding patterns matching those of the MS-H strain, the isolates that presented different patterns, and the ts+ MS-H vaccine strain (control) were all susceptible to tiamulin and tylosin and were resistant to curcumin, florfenicol, and erythromycin (Table 4). The isolates with DNA banding patterns matching those of the MG-F strain and MG-F vaccine strain (control) were susceptible to enrofloxacin, tiamulin, and tylosin but resistant to curcumin, florfenicol, and lincomycin. The isolates with DNA banding patterns matching those of untyped MG strains and the ts+ TS-11 vaccine strain (control) were sensitive to erythromycin, tiamulin, and tylosin but resistant to curcumin, enrofloxacin, florfenicol, and lincomycin (Table 5). 

## 3. Discussion

CRD and CCRD are two diseases that commonly affect poultry production in Mexico and many other parts of the world. One of the main forms of control is vaccination with live strains of MS and MG that do not grow at 39.5 °C or higher temperatures because vaccinating with strains unable to reproduce at body temperature prevents vaccine strain mycoplasmas from causing sepsis or transmitting to the egg. In this study, from laying hens previously vaccinated with the ts+ MS-H strain, we isolated MS at 39.5 °C with DNA banding patterns identical to those of the ts+ MS-H vaccine strain; therefore, in some cases, the ts+ strains were able to revert their 39.5 °C thermosensitivity and reproduce in hens. 

We found a higher rate of MS than MG isolates in both types of hens (broiler breeders and laying hens), most likely owing to the highly technical poultry industry in Mexico, as was also found in highly developed poultry farming countries in Europe and in the Americas [17,18,19,20,21,22]. In Europe, state-led MG eradication programs have been implemented [21]. In the Americas, the USA has a voluntary testing and certification program for flocks free of both mycoplasmas, while in the rest of the continent, control programs aimed at biosecurity with mycoplasma-free birds or mycoplasmosis control with antibiotic metaphylaxis are voluntary for private poultry farmers [1,23]. Additionally, in this study, we observed that MG was 5 to 10 times more sensitive than MS to two of the world’s most widely used antibiotics in the poultry industry (tiamulin and tylosin); therefore, with the use of the same dose of antibiotic treatment, the likelihood of eliminating MG is higher than that of eliminating MS.

A ts+ (39.5 °C) vaccine strain that reverses its sensitivity may reproduce in birds and cause CRD as well as transmit to the progeny. Thermosensitivity reversal in ts+ MS-H strains has been demonstrated by isolating strains typed as MS-H at 39.5 °C using restriction fragment length polymorphism analysis in broiler breeders and laying hens [5]. Under laboratory conditions, without performing molecular tests, it was shown that this thermosensitivity reversal was not a complete reversion to virulence when the ts- MS-H were isolated from lesions in specific-pathogen-free pullets (SPAFAS) [16]. In our study, using RAPD, we isolated and identified MS-H strains that grew at 39.5 °C exclusively from tracheal samples of broiler breeder and laying hens.

The samples were obtained from egg-producing hens vaccinated with the ts+ MS-H strain between two and four weeks of age, suggesting that the MS-H strain reversed thermosensitivity after 16 to 25 weeks of application. Therefore, isolated ts- MS-H strains can cause MS contamination in eggs, reducing egg production and causing economic loss. In broiler breeders, MS transmission to their progeny could cause CRD and mortality in broilers. Due to these disorders, it is essential to consider the possible risk of a reversal of pathogenicity when applying the ts+ MS-H strain or other live ts+ vaccines against *Mycoplasma* spp. In specific-pathogen-free hens vaccinated under laboratory conditions, Armour and Ferguson-Noel [24] and El Gazzar et al. [25] demonstrated that the ts+ MG TS-11 strain could revert its thermosensitivity and pathogenicity, cause septicemia, and invade the ovary, thus infecting the eggs. 

In our study, of the 12 isolated and characterized MG strains, no strain showed DNA banding patterns matching those of the ts+ MG TS-11 strain; as a result, we were unable to demonstrate that they reversed their thermosensitivity. In our study, we found three MG strains whose DNA banding patterns matched those of the F strain, which was originally a vaccine strain; currently, the F strain can be considered a vaccine or field strain, with nine strains corresponding to untyped wild-type field strains. In the study by Armour and Ferguson-Noel [24], vaccine mycoplasmas failed to colonize bird tissues; however, under field conditions, as in our study, wild strains can compete for receptors of target cells with the vaccine strain TS-11. Most likely, this competition is one of the reasons why this strain was not isolated successfully. 

In Mexico, and in many other countries, CRD is controlled with vaccines and antibiotic metaphylaxis; accordingly, the sensitivity to different antibiotics must be periodically assessed by region and by country. Our findings showed that our *Mycoplasma* isolates, in general, were of intermediate sensitivity or resistant to lincomycin, florfenicol, erythromycin, enrofloxacin, and curcumin. The *Mycoplasma* isolates, both untyped wild type and vaccine strains, were only sensitive to tiamulin and tylosin. The majority of studies on antimycoplasmic effects were performed with MG [7]; however, the MIC_100_ for MG is lower than that for MS. Thus, effective control of both *Mycoplasma* species requires using a dose 3 to 5 times higher than that needed to control MS. Regarding erythromycin, we agree with the work of Gautier-Bouchardon [26], which states that MS is inherently resistant to the antibiotic while MG is not.

In the search for new drugs that are effective against bacteria and safe for the environment as well as for humans, well-characterized products of plant origin, including polyphenols, such as curcumin, have been evaluated. Curcumin previously exhibited an antimycoplasmic effect against *Mycoplasma hominis*, *Mycoplasma capricolum*, *Mycoplasma mycoides*, *Mycoplasma genitalium*, and *Mycoplasma pneumoniae* at concentrations ranging from 50 to 100 µg mL^−1^ [11]. We considered the strains that grew at a dilution greater than 2 µg mL^−1^ as resistant to *Mycoplasma*. In this study, all MS and MG strains grew at 2.5 µg mL^−1^, which was the maximum concentration, showing that curcumin was ineffective against MS and MG at the concentrations economically profitable for poultry production.

## 4. Materials and Methods

### 4.1. Sampled Farms

The present study was conducted from January 2010 to December 2019. Samples were collected from 14 poultry companies with a medical history of CRD in their hens or chicks (eight broiler breeders and four laying hens). Samples of broiler breeders were collected from the following Mexican states: Jalisco, Querétaro, Aguascalientes, State of Mexico, Chiapas, and Veracruz. Samples of laying hens were collected from the following Mexican states: Jalisco and Puebla. 

#### Farm Selection Criteria

Broiler breeder farms whose progeny had a history of serosal-fibrinous airsacculitis or fibrinous arthritis were selected for this study, whereas the selected laying hen farms had a history of hens with airsacculitis or peritonitis and low egg production ranging from 2% to 10%. The hens of the selected farms were previously vaccinated against MS using live strain ts+ MS-H (Vaxsafe^®^ MS, Laboratorio Avimex, Mexico City, Mexico) and against MG with one of the two live strains ts+ TS-11 (TS-11^®^, Boehringer Ingelheim Vetmedica, Guadalajara, Mexico) or strain F (F VAX-MG^®^, MSD Animal Health, Mexico City, Mexico).

### 4.2. Bacterial Isolation

In total, 10 hens were randomly selected from each farm. Tracheal scrapping samples were collected using a swab and transported in FREY liquid medium. The samples were incubated at 39.5 °C for 20 days [27,28]. Cultures were reviewed daily, and subcultures of samples showing changes in the pH were prepared according to the method described by Poveda [29]. The cultures that showed no change in pH were considered negative. MS and MG were identified in biochemical tests (glucose fermentation, tetrazolium reduction, arginine hydrolysis, and digitonin sensitivity), and the results were confirmed with direct immunofluorescence testing. Cloning was completed by aspirating single colonies with a Pasteur pipette and inoculating broth medium to eliminate clumped cells and then re-plating on agar. The cultures were cloned three times to ensure purity. After sufficient growth in broth, the medium was filtered through a 0.45 µM filter to eliminate clumped cells and re-plated on agar, as described by Poveda [29] and Ferguson-Noel and Kleven [30]. To serve as controls, original strains of the MS-H and TS-11 vaccines were incubated at 33 °C, and the F strain was incubated at 37 °C.

### 4.3. Molecular Identification of Mycoplasma spp. Vaccine Strains

The DNA of isolated and cloned MS and MG strains was identified by RAPD using the technique and primers designed by Geary et al. [31]. The DNA banding patterns of the isolated strains were compared to the DNA banding patterns of the MS-H, MG-F, and MG TS-11 reference strains. 

### 4.4. In Vitro Sensitivity to Antibiotics and Curcumin

MIC_100_ of six antibiotics and curcumin (1, C_21_H_20_O_6_) was calculated with the isolated *Mycoplasma* strains. The antibiotics evaluated in this study were selected based on widespread usage in poultry in Mexico for mycoplasma control and classification as critically important for both veterinary and human use by the World Health Organization [32]. The following antibiotics were obtained from Merck (Kenilworth, NJ, USA): lincomycin hydrochloride, florfenicol, erythromycin thiocyanate, enrofloxacin, tiamulin hydrogen fumarate, and tylosin tartrate. Polyphenol 4.5% curcumin (Laboratorios Mixim, Public Limited Company, Naucalpan, Mexico) was chemically dispersed into polyvinyl pyrollidone (Plastone K-29/32, Ashland™, Mexico City, Mexico). The dilutions were calculated in relation to the final concentration of the base molecule. MIC_100_ values were defined as the lowest concentration of the antibiotic at which 100% of the isolates were inhibited. 

All isolated strains were assayed for antibiotic MIC_100_, whereas only five strains from each *Mycoplasma* were evaluated against curcumin. MIC_100_ was calculated according to the method described by Hannan [33] using FREY culture medium [27,29]. Following the recommendation of Hannan [33], the culture concentration of each isolated *Mycoplasma* spp. was adjusted to 10^4^ color changing units mL^−1^, with phenol red as the indicator. Two-fold dilutions of the antibiotic were prepared, ranging from 2.5 to 0.01 µg mL^−1^, although there are no official cut-off points for the interpretation of MIC_100_ for avian mycoplasma [26,33]. In this investigation, isolates were considered susceptible to antibiotics when the MIC_100_ was 0.5 g mL^−1^. Isolates with MIC_100_ 1 g mL^−1^ were classified as intermediate to antibiotics and those with MIC_100_ 2 g mL^−1^ were classified as resistant. These criteria were adapted from the research of Lysnyansky et al. [34].

### 4.5. Statistical Analysis

The proportion of the number of isolates was compared using the chi-squared test. The mean MIC_100_ of each antibiotic was compared between MS and MG isolates using Student’s *t*-test. An analysis of variance (ANOVA) was performed on the MIC_100_ results of the same antibiotic from typified strains (based on DNA banding pattern), and the differences between the means were evaluated using Tukey’s honestly significant difference (HSD) test. Statistical significance was set at *p* < 0.05.

### 4.6. Ethical Compliance

This study was conducted in accordance with the recommendations of the Institutional Animal Care and Use Committee (IACUC) at the University of Arkansas (Fayetteville, AR, USA) under approved protocol #17073.

## 5. Conclusions

Based on the findings of this study, we propose that MS is the most important *Mycoplasma* for Mexican poultry production for the following reasons: MS was the *Mycoplasma* species most frequently isolated, the vaccine used (ts+ MS-H) was able to revert its temperature sensitivity and recover virulence, and MS was 5 to 10 times less sensitive than MG to the most widely used antibiotics (tiamulin and tylosin). Further studies evaluating the virulence of these thermosensitive revertant strains on egg production and on the progeny of hens vaccinated with MS-H are necessary to determine the economic and health impact of this reversal. 

## Figures and Tables

**Table 1 pathogens-09-00924-t001:** *Mycoplasma synoviae* (MS) and *Mycoplasma gallisepticum* (MG) isolates in lots of broiler breeders and laying hens in different states of Mexico.

	Broiler Breeders							Laying Hens			Total Hens
	Ags	Chis	SoM	Jal	Qro	Ver	Total	Jal	Pue	Total	
Sampled lots	6	52	102	90	84	57	391	37	203	240	631
*M. synoviae*	2	15	16	26	9	32	100	7	16	23	123
*M. gallisepticum*	2	0	14	0	6	1	23	0	0	0	23

Ags = Aguascalientes; Chis = Chiapas; SoM = State of Mexico, Jal = Jalisco, Qro = Querétaro, Ver = Veracruz, and Pue = Puebla.

**Table 2 pathogens-09-00924-t002:** The RAPD identification of *Mycoplasma synoviae* and *Mycoplasma gallisepticum* vaccine strains isolated at 39.5 °C in broiler breeders and laying hens in different states of Mexico.

	Ags	Chi	SoM	Jal	Pue	Qro	Ver	Total
MS-H strain	2	5	--	5	6	1	5	24
Untyped *M. synoviae*	0	0	--	3	1	0	0	4
MG-F strain	2	--	0	--	--	--	1	3
Untyped *M. gallisepticum*	0	--	9	--	--	--	9	9

RAPD = Random amplification of polymorphic DNA. Ags = Aguascalientes; Chi = Chiapas; SoM = State of Mexico, Jal = Jalisco, Pue = Puebla, Qro = Querétaro, and Ver = Veracruz.

**Table 3 pathogens-09-00924-t003:** The mean ± standard deviation in µg mL^−1^ of the minimum inhibitory concentration (MIC_100_) of *Mycoplasma synoviae* and *M. gallisepticum* isolates from broiler breeders and laying hens in Mexico.

	Curcumin *	Enrofloxacin	Erythromycin	Florfenicol	Lincomycin	Tiamulin	Tylosin
*M. synoviae*	≥2.50 ± 0.00 ^a^(R)	1.64 ± 0.74 ^a^(I)	≥2.50 ± 0.00 ^b^(R)	2.23 ± 0.52 ^a^(R)	1.81 ± 0.66 ^a^(I)	0.43 ± 0.30 ^b^(S)	0.92 ± 0.41 ^b^(I)
*M. gallisepticum*	≥2.50 ± 0.00 ^a^(R)	1.90 ± 1.09 ^a^(I)	0.48 ± 0.78 ^a^(S)	≥2.50 ± 0.00 ^b^(R)	≥2.50 ± 0.00 ^b^(R)	0.03 ± 0.02 ^a^(S)	0.07 ± 0.05 ^a^(S)

* Five isolates of each *Mycoplasma* species were used for curcumin. Different letters between the same antibiotic indicate significant differences (*p* < 0.05). R = resistant isolates, I = isolates with intermediate sensitivity, and S = sensitive isolates.

**Table 4 pathogens-09-00924-t004:** The means ± standard deviation in µg mL^−1^ of the minimum inhibitory concentration (MIC_100_) of *Mycoplasma synoviae* isolates by DNA banding pattern in hens in Mexico.

	Curcumin	Enrofloxacin	Erythromycin	Florfenicol	Lincomycin	Tiamulin	Tylosin
ts- MS-H	≥2.50 ± 0.00 ^a^(R)	1.72 ± 0.70 ^b^(I)	≥2.50 ± 0.00 ^a^(R)	2.19 ± 0.55 ^a^(R)	1.80 ± 0.67 ^b^(I)	0.45 ± 0.31 ^a^(S)	0.95 ± 0.41 ^b^(S)
Untyped *M. synoviae*	≥2.50 ± 0.00 ^a^(R)	1.17 ± 0.97 ^b^(I)	≥2.50 ± 0.00 ^a^(R)	≥2.50 ± 0.00 ^a^(R)	1.88 ± 0.72 ^b^(I)	0.31 ± 0.22 ^a^(S)	0.70 ± 0.39 ^b^(S)
ts+ MS-H	≥2.50 ± 0.00 ^a^(R)	0.31 ± 0.00 ^a^(S)	≥2.50 ± 0.00 ^a^(R)	≥2.50 ± 0.00 ^a^(R)	0.63 ± 0.00 ^a^(S)	0.08 ± 0.00 ^a^(S)	0.02 ± 0.00 ^a^(S)

Five isolates of each *Mycoplasma* species were used for curcumin. Different letters between the same antibiotic indicate significant differences (*p* < 0.05). R= resistant isolates, I = isolates with intermediate sensitivity, and S = sensitive isolates.

**Table 5 pathogens-09-00924-t005:** The means ± standard deviation in µg mL^−1^ of the minimum inhibitory concentration (MIC_100_) of *Mycoplasma gallisepticum* isolates by DNA banding pattern in hens in Mexico.

	Curcumin	Enrofloxacin	Erythromycin	Florfenicol	Lincomycin	Tiamulin	Tylosin
MG-F strain	≥2.50 ± 0.00 ^a^ (R)	0.08 ± 0.00 ^a^(S)	1.67 ± 0.72 ^a^(I)	≥2.50 ± 0.00 ^b^ (R)	≥2.50 ± 0.00 ^a^ (R)	0.05 ± 0.02 ^a^(S)	0.04 ± 0.00 ^a^(S)
Untyped *M. gallisepticum*	≥2.50 ± 0.00 ^a^ (R)	≥2.50 ± 0.00 ^c^(R)	0.08 ± 0.05 ^b^(S)	≥2.50 ± 0.00 ^b^ (R)	≥2.50 ± 0.00 ^a^ (R)	0.02 ± 0.01 ^a^(S)	0.08 ± 0.05 ^a^(S)
MG-F strain	≥2.50 ± 0.00 ^a^(R)	0.08 ± 0.00 ^a^(S)	0.01 ± 0.00 ^c^(S)	≥2.50 ± 0.00 ^b^ (R)	≥2.50 ± 0.00 ^a^ (R)	0.01 ± 0.00 ^a^(S)	0.01 ± 0.00 ^a^(S)
ts+ MG TS-11	≥2.50 ± 0.00 ^a^(R)	0.39 ± 0.00 ^b^(S)	0.01 ± 0.00 ^c^(S)	0.62 ± 0.00 ^a^(S)	≥2.50 ± 0.00 ^a^(R)	0.01 ± 0.00 ^a^(S)	0.01 ± 0.00 ^a^(S)

Five isolates of each *Mycoplasma* species were used for curcumin. Different letters between the same antibiotic indicate significant differences (*p* < 0.05). R= resistant isolates, I = isolates with intermediate sensitivity, and S = sensitive isolates.

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
