# Peer review of "Isolation and Antimicrobial Sensitivity of Mycoplasma synoviae and Mycoplasma gallisepticum from Vaccinated Hens in Mexico"

_pathogens, 2020, doi:10.3390/pathogens9110924_

Round 1

Reviewer 1 Report

Although interesting, manuscript needs to be improved prior to publication. In particular, authors should clearly specify whether detected MIC values are MIC100. Moreover, since the significance of the manuscript is all based on the antimicrobial efficacy of natural compounds, further experiments should be performed to verify whether effective compounds and conditions are bacteriostatic or bactericidal.

Author Response

Dear Reviewer, we greatly appreciate your suggestions, we try to improve the text with your recommendations; especially regarding methods, where it was clarified that the MIC is MIC100. As far as curcumin is concerned, we tested its effectiveness up to 150 µg mL-1 and we did not observe any bacteriostatic or bactericidal activity against M. synoviae or M. gallisepticum.  Furthermore, we have decided to use the MDPI English editing service, so our manuscript is checked for grammar, spelling, punctuation, and some improvement of style where necessary.  Thank you.

Reviewer 2 Report

The submitted manuscript provides a very significant information to the poultry farming and industry. The identification of vaccine reversion to become virulent and infective to the hen and eggs raise an important issue of contamination and economic lost. Though the emphasis of the data is lightly mentioned in Line 145, I would urge the authors to make it as an alert to future use of such thermo-unstable vaccines.

Few minor comments found in the manuscript:

  1. The topic could make a correction as "Isolation............... from hens in Mexico".
  2. Line 25, Should it be MG TS-11 or MG F strains?
  3. Line 55, Sentence need to be revised. "safe med for environment?
  4. Line 74, Sentence need to be revised. "This In total? Please start the sentence more appropriately.

Author Response

Dear Reviewer, thank you very much for the time you have spent on reviewing our manuscript your comments are very valuable and helpful for revising our paper and guiding our researches. We have studied those comments carefully and have made corrections, which we hope meet with the approval. Revised portion in the new version were included using the Word tracking feature in the paper.  Furthermore, we have decided to use the MDPI English editing service, so our manuscript is checked for grammar, spelling, punctuation, and some improvement of style where necessary.  Thank you.

The following is our point-by-point response to reviewers’ comments:

Few minor comments found in the manuscript:

  1. The topic could make a correction as "Isolation............... from hens in Mexico".

Suggestion accepted.  The title has been modified.  Thank you.

  1. Line 25, Should it be MG TS-11 or MG F strains?

No, it refers to MS-H and MG F strains, thank you.

  1. Line 55, Sentence need to be revised. "safe med for environment?

Suggestion accepted, thank you.

  1. Line 74, Sentence need to be revised. "This In total? Please start the sentence more appropriately.

Suggestion accepted, thank you.

Round 2

Reviewer 1 Report

Manuscript is now suitable for publication.